# The Impact of Membrane Protein Diffusion on GPCR Signaling

**DOI:** 10.3390/cells11101660

**Published:** 2022-05-17

**Authors:** Horst-Holger Boltz, Alexei Sirbu, Nina Stelzer, Primal de Lanerolle, Stefanie Winkelmann, Paolo Annibale

**Affiliations:** 1Zuse Institute Berlin, Takustraße 7, 14195 Berlin, Germany; winkelmann@zib.de; 2Department of Mathematics and Computer Science, Free University Berlin, Arnimallee 14, 14195 Berlin, Germany; 3Max Delbrück Center for Molecular Medicine, Robert-Rössle-Straße 10, 13125 Berlin, Germany; alexei.sirbu@mdc-berlin.de (A.S.); nina.stelzer@bih-charite.de (N.S.); 4Department of Physiology and Biophysics, College of Medicine, University of Illinois at Chicago, 835 S. Wolcott, Chicago, IL 60612, USA; primal@uic.edu; 5School of Physics and Astronomy, University of St. Andrews, North Haugh, St. Andrews KY16 9SS, UK

**Keywords:** GPCR signaling, spatiotemporal signal shaping, diffusion-limited reaction

## Abstract

Spatiotemporal signal shaping in G protein-coupled receptor (GPCR) signaling is now a well-established and accepted notion to explain how signaling specificity can be achieved by a superfamily sharing only a handful of downstream second messengers. Dozens of G${}_{s}$-coupled GPCR signals ultimately converge on the production of cAMP, a ubiquitous second messenger. This idea is almost always framed in terms of local concentrations, the differences in which are maintained by means of spatial separation. However, given the dynamic nature of the reaction-diffusion processes at hand, the dynamics, in particular the local diffusional properties of the receptors and their cognate G proteins, are also important. By combining some first principle considerations, simulated data, and experimental data of the receptors diffusing on the membranes of living cells, we offer a short perspective on the modulatory role of local membrane diffusion in regulating GPCR-mediated cell signaling. Our analysis points to a diffusion-limited regime where the effective production rate of activated G protein scales linearly with the receptor–G protein complex’s relative diffusion rate and to an interesting role played by the membrane geometry in modulating the efficiency of coupling.

## 1. Introduction

G protein-coupled receptors (GPCRs) constitute an important [1] and large class of proteins mediating the cellular response to specific external stimuli. Functionally, it is vital for the cell to modulate the relative efficacy of signals originating from distinct GPCRs [2,3]. Any means of modulation of the local signaling machinery can, in principle, lead to downstream spatiotemporal shaping of the signal when combined with other localization mechanisms, such as buffered diffusion of second messenger molecules [4], commonly referred to as compartmentalization. In general, the malleability of the reaction-diffusion processes involved in signaling is a way to overcome the relatively low variety in downstream pathways [5,6] and accurately reflect the complex abundance of potential stimuli. Several mechanisms of signal shaping have been observed, many of which concern the downstream signaling cascade that is triggered by the receptor interacting with a ligand and a heterotrimeric G protein [7].

GPCR signaling is, at its heart, a ternary binding process. The receptor molecule (membrane-bound) binds to a ligand (coming from the cell’s exterior), leading to an increased affinity to bind to a G protein (membrane-bound, coming from within the cell membrane) [8]. A guanine nucleotide exchange (GDP-to-GTP exchange) then activates the G protein. G proteins are divided into subclasses according to the downstream signaling activity that they elicit. While the general mechanism holds for multiple signaling pathways, our presentation focuses on the G${}_{s}$ case, as this is the prototypical example [9,10]. Heterotrimeric G proteins have three subunits, labeled α, β and γ, which for the purpose of GPCR signaling effectively constitute a dimer, with the signal being transduced by either α or βγ. Specifically, G$\alpha_{s}$ stimulates adenylate cyclase and cAMP production upon activation. A simplified reaction network scheme is given in Figure 1.

We note that only the very first segments in this sequence of events are influenced by the receptor’s single-particle diffusive properties. Receptor binding to the G protein consists of two steps that we can conceptually depict as (1) a search phase, in which the binding proteins get sufficiently close to each other, and (2) the actual interaction. The search process is a relevant contribution because energy is consumed in the signaling cascade, manifesting itself by replacing a GDP in the Gα with a GTP. Eventually, if the reverse reaction were not carried over (by hydrolizing GTP to GDP), the pool of available G proteins would deplete. The local depletion of available and (for the signaling) usable G protein induces an inhomogeneity in the system that would invalidate the assumption of being at equilibrium (or well-stirred), underlying the usual mass action law kinetics. The travel times contribute to every reaction and, as discussed in the following, may dominate the inter-reaction times. In terms of physical modeling, energy consumption during the GDP-to-GTP exchange introduces a manifest irreversibility at this stage that has to be accounted for. Thus, the binding of the receptor–ligand complex to the G protein is conceptually different from the one of the receptor and ligand. The latter can and will rebind frequently, and the overall effect of a change in receptor dynamics is comparatively small due to large differences in molecular weight, with typical ligands being roughly one (large peptidic ligands) to three (adrenergic ligands) orders of magnitude smaller than their respective receptors. Additionally, the ligands’ motion outside the cell occurs in three dimensions, and any search process is substantially faster. Therefore, we focus our survey on the binding of a receptor–ligand complex to the G protein and the formation of an active ternary complex.

As the temporal order of binding events to form the ternary complex matters, there are multiple possible scenarios in this regard [11]. For simplicity and without affecting the generality of our conclusions, we focus here on the case of a preformed receptor–ligand complex binding with a G protein. This is because the activated receptor state has an increased lifetime [12,13] as well as lower energy barriers [14]. We therefore disregard the case of the receptor binding directly to the G protein without a ligand.

In this perspective, we emphasize the relevance of receptor (or receptor–ligand complex) dynamics as a mechanism that cells may employ to modulate the signaling machinery, and thus specificity, in the downstream pathway. The underlying core insight is that the binding reaction of the receptors or receptor–ligand complexes to the G protein is diffusion-limited [15,16,17,18], meaning that inter-reaction times are dominated by the time needed for the binding partners to meet. As a consequence, single-particle dynamics are relevant in a way that is easily overlooked in standard mass action law approaches. We review the relevant ideas in the literature with the goal of an accessible biophysical presentation that focuses on this particular aspect of the signaling machinery to address the leading question: how do receptor and G protein dynamics at the plasma membrane affect GPCR signaling? We will also discuss the outline of how this could be addressed experimentally together with the presentation of some relevant data to support the discussion.

## 2. Diffusion Limitation in GPCR-G Protein Binding

In the following, we analyze the scaling of the effective production rate of activated G protein, depending on the diffusion regime, separating between fast and slow diffusion on the plasma membrane. We start in Section 2.1 by introducing a reduced model for the diffusion and binding dynamics of a single receptor–ligand complex and a G protein. In Section 2.2, we analyze the basic scenario of diffusion in a planar membrane, while in Section 2.3, more complicated geometries as well as clustering and oligomerization effects are considered.

### 2.1. Reduced Diffusional Model

A receptor can form several complexes with external ligands as well as those with the cognate G protein, all of which can determine conformational changes in the receptor. This has led to the description of the reaction kinetics by means of the *cubic ternary complex model* or extensions thereof [12]. Similarly, other models have been used to capture other aspects of the GPCR signaling [19,20]. Here, we want to present a very concise model that is focused on the effect of receptor diffusion dynamics and allows for direct analytical (in terms of dimensional scaling arguments) and numerical investigation. Of particular interest are the deviations from a simple mass action reaction law that follow from the local depletion of G protein. Following a reaction and its activation at a given receptor, the G protein is not directly available again at the receptor but is, on the population level, replenished elsewhere. This provides intuitive insight into the role that diffusion limitation plays in the GPCR signaling cascade. Furthermore, due to its simplicity, we are directly able to extent the model to other geometries than the planar membrane. The interested reader may find further references, either characterized by more analytical [21,22,23,24,25] or more biological [15,26,27] approaches, that lead to similar conclusions.

We take a black box approach to those steps in the GPCR signaling process that do not directly depend on the receptor diffusion constant and only model the receptor–ligand complex (RS) and G protein (G) binding. This is a choice, and one may consider the case of “pre-coupled” receptors (receptors binding to G protein before binding to a ligand) as well. We take a single-molecule level approach along the lines of the Doi model [28], a standard tool in particle-based reaction-diffusion modeling, accounting for inter-molecular interactions of diffusing molecules in terms of an interaction range ℓ>0 and a reaction rate k>0.

The result of our considerations on this model is an *effective rate* $k_{\mathrm{eff}}$, describing the rate of production of activated G protein. At the cell level, the concentration of activated G protein controls the resulting signaling strength. As mentioned before, there are conceptually two contributions to this effective rate: (1) searching an available G protein (i.e., getting within a distance smaller than *ℓ*) and (2) the actual binding (which happens with a constant rate *k* while within interaction range). The length scale *L* associated with the typical distance to the available G protein should be large compared with the interaction range (i.e., L≫ℓ). As a consequence of this separation of scales, and to reduce model complexity, we take an effective single-particle approach: we consider the binding of a single receptor–ligand complex to a single G protein in the frame of reference of the G protein such that there is a static interaction zone and an effective diffusion constant $D=D_{R}+D_{G}$. This starts at a distance *L* to the interaction zone and is a renewal process that is restarted upon binding. The actual geometry and distribution of the distances are neglected, as we primarily focus on the scaling, particularly with the diffusion coefficient *D*. From the first reaction time τ (i.e., the time it takes the process to reset), we can extract the effective rate of activated G protein production via
$$\begin{array}{c} k_{\mathrm{eff}}=1/\langle \tau \rangle , \end{array}$$
where 〈τ〉 refers to the mean of τ.

In summary, our model is sketched in Figure 2. A particle diffuses in a circular domain of radius *L* with a diffusion constant *D*. Inside, there is a smaller concentric circular region of radius *ℓ*, to which the particle can bind with a constant k>0 while inside. Where necessary, we assume reflecting boundary conditions.

### 2.2. Scaling Analysis

We can gather some initial insight from dimensional analysis of the model parameters, particularly from considering the timescales present in this model. Trivially, binding with a rate of *k* implies the typical timescale of binding: $$\begin{array}{cc} \tau_{k} & ∼\frac{1}{k}. \end{array}$$

As binding is only relevant while the particle is inside the inner region, we have to compare $\tau_{k}$ to the timescale corresponding to the typical time spent inside this region, which is given by
$$\begin{array}{cc} \tau_{\ell } & ∼\frac{\ell^{2}}{D}, \end{array}$$
and to the time spent to initially reach this region: $$\begin{array}{cc} \tau_{L} & ∼\frac{{\left(L-\ell \right)}^{2}}{D}. \end{array}$$

We are mainly interested in a regime where $\tau_{L}>\tau_{\ell }$ as L≫ℓ, due to typical G protein plasma membrane densities of $\left[G\right]\approx 10^{5-6}/\mathrm{cell}$. Assuming a surface area of the cell to the order of 1000 (μm)${}^{2}$, this leads to [G]≈ 10${}^{2-3}$/(μm)${}^{2}$ or $L^{-1}\approx$ 10${}^{1-2}$ (μm)${}^{-1}$, which is larger than the interaction ranges of ℓ⪅nm [26]. Dimensionally, we expect a crossover between different behaviors around a critical diffusion constant $D_{c}$, for which the two timescales $\tau_{L}$ and $\tau_{k}$ are of the same order of magnitude: $$\begin{array}{c} \tau_{k}/\tau_{L}∼1\;\Rightarrow \;D_{c}∼{\left(L-\ell \right)}^{2}k\approx L^{2}k. \end{array}$$

It is instructive to use this scale in order to consider the extreme cases of very small and very large diffusivity or, conversely, large and small binding rates.

In the case where $D\gg D_{c}$, the time typically spent in the binding area is small compared with the binding time (i.e., there will be many passages through this area before the particle actually binds and a negligible contribution from the initial search). From this, we find that the effective rate should scale like the equilibrium probability to be in the reaction area in the nonbinding case. Without any binding interaction, the equilibrium distribution is uniform, with every point within the radius of size *L* being equally likely. Considering a virtual ensemble of such systems, a fraction corresponding to the ratio of areas $P_{\mathrm{inside}}\propto {\left(\ell /L\right)}^{2}$ would be inside the interaction range. The interaction rate will be directly proportional to this while also being proportional to the original binding rate: $$\begin{array}{cc} k_{\mathrm{eff},D\gg D_{c}} & ∼{\left(\ell /L\right)}^{2}k. \end{array}$$

Essentially, this fast diffusion regime corresponds to the typical assumption of a well-stirred state underlying the mass action law. Thus, if the diffusion is high compared with the threshold value $D_{c}=L^{2}k$, we do not expect a relevant effect from the receptor diffusion dynamics to the signaling.

In the case where $D\ll D_{c}$, the actual binding is fast compared with the travel times. Spatial heterogeneity matters, and the travel times become the determining factor for the effective binding rate, leading to
(1)$$\begin{array}{cc} k_{\mathrm{eff},D\ll D_{c}} & ∼\frac{D}{{\left(L-\ell \right)}^{2}}\approx \frac{D}{L^{2}}. \end{array}$$

In other words, the effective production rate of activated G protein scales linearly with the diffusion coefficient.

In order to determine in which regime actual cells are operating, we need to review here some of the ranges currently encountered in the literature for such values. Concerning diffusion constants on the plasma membrane, a consensus value is typically found in the range of of D∼ 0.1 μm${}^{2}$/s [29,30,31,32]. There is some level of experimental control that can be enacted over the diffusion constant. One approach is to change the membrane composition [33,34], which affects the viscosity μ of the membrane for the receptor molecules and, by way of the Einstein–Smoluchowski (or fluctuation–dissipation) relation, the diffusion constant as $D∼\mu^{-1}$. Similarly, GPCR interacting proteins (GIPs), transmembrane or cytosolic proteins that can bind to both the heterotrimeric G protein and the receptor, can affect the receptor dynamics and have been implicated in fine-tuning signaling [35]. Another approach to control the receptor dynamics is by means of modulating the actin expression [36]. Actin interacts with the receptors by providing a background of specific as well as unspecific interactions that will effectively slow down the diffusion [37], as clearly observed, for example, for the transferrin receptor (a non-GPCR transmembrane protein) [38]. Thus, an increase in polymerized actin should lead to a decrease in the observed diffusion constant, whereas a reduction in the actin fiber density should lead to an increase. This intuitive notion has been confirmed experimentally by us and others for the case of overexpression [36] (see also Figure 3). Additionally, the relative importance of free and polymerized actin can be changed by introducing mutations (such as the S14C) which facilitate polymerization [39]. Drugs that induce depolymerization of actin, such as Latrunculin B (LatB), lead to a loss of actin fibers to the point of a collapse of the cortex [40]. The effect of overexpression is found to be independent of the local actin network topology, hinting to it not being a mere scaffolding effect [36].

On the other hand, the actual values for the length scale *L* and reaction rate *k* are subject to more variability in the available literature, ranging from low values of k∼ 1/s [12,13] to high values of k∼$10^{3}$/s [41,42]. This wide range in reported values is interesting in itself, with the temporal trend being toward lower reported values of *k* in the last few years. A more definitive consensus for appropriate effective rates for modeling purposes would be a very welcome addition to the literature. We note that our parameter *k* conceptually only captures the binding, and the actual activation times can and will be larger than the binding times. The length scale *L* is modulated by the expression levels of the signaling partners, which may be cell type-dependent. As stated earlier, a rough figure is given by $L^{-1}\approx 10^{1-2}$ (μm)${}^{-1}$. If we use the available experimental data from [26], we can estimate that the times between interactions are about an order of magnitude greater than the time needed for the activation of G protein once in contact with a receptor, which has been found to be several milliseconds [42]. In our terminology, this means $\tau_{k}\ll \tau_{L}$ or $D\ll D_{c}$, and we are in the slow diffusion or diffusion-limited regime, expecting an effective rate of $k_{\mathrm{eff}}\propto D$ that is strongly dependent on the diffusion constant and therefore allows for immediate modulation by any change in receptor dynamics. Covering a wide range of parameter values is appropriate, especially in case of the reaction rate *k*, as there are reported values of *k* that imply reaction times comparable to or larger than the diffusion times [43]. In this case, $k_{\mathrm{eff}}∼D$ no longer holds, and the dependence of the signaling strength upon the diffusion constant is reduced, albeit still present. To account for this, we keep our discussion below about the implications on the reaction network in terms of the effective rate $k_{\mathrm{eff}}$.

The scaling result can be immediately appreciated in our graphical display of Figure 4. Here, we present the results of a straightforward simulation of our model (the details of which are given in Appendix A) and show the empirically found effective rates as functions of the diffusion constant. The simulation uses dimensionless units, but for the sake of convenience, we also display the results in physical units (top axis), assuming values of $k=10^{2}$ s${}^{-1}$ and $\ell^{2}=10^{-15}$ m. The relatively large interaction range is motivated by the fact that simulation times become very long for L/ℓ≫1, and we can get to larger travel distances L−ℓ with the same computational resources this way. We can thus corroborate our scaling analysis with a crossover from a strongly diffusion-dependent regime for small values of the diffusion coefficient *D* to a regime where diffusion becomes irrelevant, and the bare binding rate reappears.

### 2.3. More Complicated Scenarios: Clustering and Geometries Other Than Planar

While the presentation in terms of the Doi model may appear simplistic, the diffusion-limited result k∼D does also hold in more involved descriptions. One conceptually interesting refinement would be the effect of clustering, an important manifestation of this being oligomerization of the receptor molecules [44,45,46,47,48,49]. Clustering can be somewhat captured in our models by effective parameters. A cluster of targets will increase not only the adsorption rate *k* and the size *ℓ* of the target, but the spatial heterogeneity as captured by the typical distance *L* will also increase. Moreover, the diffusion dependence will not only prevail [50], but when assuming *D* to be constant, this change in parameters would imply an increase in $D_{c}=kL^{2}$ and therefore move the system deeper into the diffusion-limited regime (corresponding to moving down and left in Figure 4 (left)). Another effect of clustering, however, is that it changes the relevance of lifetime effects. We do not account for the unbinding and rebinding of receptor–ligand complexes, which are greatly altered in the presence of receptor clusters. To an extent, these effectively modulate the abundances of the relevant complexes.

Importantly, the diffusion-limited regime allows also for other ways of modulating the effective rate than directly changing the diffusion constant. This can be motivated by inspection of the free diffusion law $\delta^{2}=2dDt$, with $\delta^{2}$ being the mean squared displacement, *D* the diffusion constant, *t* the time and *d* the spatial dimensionality. While the membrane, as a lipid bilayer, is always effectively two-dimensional, introducing curvature allows for a manipulation of the relative importance of the two dimensions. It was observed experimentally that there is a modulation of the receptor concentration in curved geometries [32,51], but there is no clear evidence available for the dynamics and signaling. The simplest model one can devise to address this problem is that of a long cylinder, as the axial direction is extended while the circumferential direction is finite. This means that as particles diffuse along the cylinder, the shape of the diffusive front changes from a circle that wraps around the cylinder and ultimately merges with itself to two straight lines propagating forward. In other words, there is a crossover from two-dimensional to effectively one-dimensional diffusion on a timescale $\tau_{\rho }∼\rho^{2}/L$, determined by the cylinder radius ρ. This boils down to the effective rate $k_{\mathrm{eff}}$ being dependent on the cylinder radius ρ as well. Additionally, the manifest anisotropy of a cylinder (at larger scales) becomes important for the typical distances between a receptor and a G protein at identical densities when compared with a flat membrane. This is because the typical free area is also anisotropic in this case, leading to search times that are dominated by the longest dimension. Numerical data to verify this are given in Figure 5, where we show the first passage time from the edge of a patch with an area $\pi L^{2}$ to a central circular area of a size $\pi \ell^{2}$. To account for the area doubly covered due to the periodicity, the size of this patch is $L^{\prime }\geq L$. As a cylinder can be unrolled to a flat sheet (using the axial component and the rotational angle as coordinates), a cylinder of a sufficiently large radius is, for our purposes, indistinguishable from a flat membrane. (The curvature would, in principle, lead to an effective value of the interaction range *ℓ* if the interaction range is defined by the Euclidean distance. However, the results in the diffusion-limited regime are fairly insensitive to *ℓ*). For very thin cylinders, on the other hand, the geometry becomes important with drastically increasing first passage times in the rescaled system. Conversely, this would correspond to a decrease in the effective rate. This could be one contributing factor to the observed differences regarding signaling in these geometries, with one example being the T-tubular network of cardiomyocytes [32,52].

In more general scenarios, the geometry and therefore the dynamics will be inhomogeneous, and general statements in this respect are rare. It is important to consider the effective diffusion dynamics on a surface that is given in the so-called Monge parametrization (i.e., a point on the surface is given by a vector of the form (x,y,ϵH(x,y)), with *x*, *y* being regular Cartesian coordinates and H(x,y) some smooth function). The amplitude of the undulations can be controlled via the parameter ϵ>0. When evaluating the appropriate Laplace–Beltrami operator (which replaces the standard Laplacian in the diffusion equation when going from a planar setting to a more general surface) in a leading non-trivial order in ϵ, one finds that, when observed in the *x*–*y* plane, there is an inhomogeneous diffusion, but more interestingly, there is also an effective potential $V=\epsilon^{2}\;/\;2D\left(H_{x}^{2}+H_{y}^{2}\right)$. The effective forces associated with this are given by derivatives corroborating the intuitive notion that the curvature of the membrane modulates the dynamics. This, in turn, will affect the reaction kinetics. Some recent progress in this direction has been made by studying the effect of a small radially symmetric deviation from the planar state [53]. In a biological setting, this effect will compete with any potential energetic effect induced by a curved membrane as well as more indirect effects that modulate the binding rates (with the change in conformation of the proteins being one). We believe this is one potential avenue for future modeling, should relevant data become available.

Finally, we note that due to the complex nature of the membrane environment, in particular the crowding of other molecules [54] as well as micro-confinement, anomalous diffusion [55,56] might be relevant for short time scales (i.e., mean square displacements might follow a law $\delta^{2}∼D_{\alpha }t^{2\alpha }$ with α≠1/2 with a more generalized diffusion constant $D_{\alpha }$). In this case, we would have to adapt our reasoning by using $\tau_{L}∼{\left(D_{\alpha }\right)}^{1\;/\;2\alpha }$ [57]. The biologically relevant case is subdiffusive behavior α<1/2, although superdiffusive behavior α>1/2 is also possible due to activity [58] such as directed motion along cytoskeletal filaments that is driven by molecular motors [59]. To investigate the practical effect of such anomalous behavior qualitatively, let us assume that in an experiment, the empirical diffusion constant would be measured based on the mean squared displacements $\delta_{D}$ at some time $t_{D}$ using $D_{\mathrm{emp}}=\delta_{D}^{2}/\left(4t_{D}\right)$ (i.e., assuming $\alpha =\frac{1}{2}$), implying $4t_{D}k_{\mathrm{eff}}∼{\left(\delta_{D}/L\right)}^{2}$ (see Equation (Equation 1)). If the true dynamics are anomalous, this means we actually should use $D_{\alpha ,\mathrm{emp}}=\delta_{D}^{2}/\left(4t_{D}^{2\alpha }\right)$ and will get $4t_{d}k_{\alpha ,\mathrm{eff}}∼{\left(\delta_{D}/L\right)}^{1/\alpha }$. Therefore, a “freely diffusive expectation” based on large times such that $\delta_{D}>L$ would underestimate the effective rate in an anomalously diffusive system with $\alpha <\frac{1}{2}$, and one based on short times, for which $\delta_{D}<L$, would overestimate it. This is an important aspect to keep in mind. For the systems of interest here, there is indeed evidence for diffusive behavior of the receptors on the timescales relevant for receptor–G protein meetings (to the order of 100 ms [26], corresponding to distances of about 200 nm using a typical diffusion coefficient of D≈ 0.1 (μm)${}^{2}$/s) [32,38].

In conclusion, a diffusion-limited reaction on a more complex geometry is not only dependent on the receptor dynamics themselves but also on their interplay with the geometry, giving ample degrees of freedom for a modulation of the reaction by the local biophysical environment of the cell.

## 3. Measuring the Second Messenger Concentration: The Reaction Network

Let us now assume that the cell, using one of the above-mentioned mechanisms, has been able to modulate the relative diffusion coefficient of the GPCR–G protein pair. We aim to determine if this translates into a modulation of the second messenger production, either locally or globally, depending on the specific geometry. Second messengers are the molecules produced in reactions that are downstream in the cascade of events triggered by the activation of a G protein. In the following, we present an abridged analysis of this downstream network that, while not suitable for a quantitative prediction, does allow for an analytical expectation for the functional gestalt of dose–response curves in experiments or its dependence on the dynamics.

### 3.1. The Signaling Network

To understand the observable effect of a change in the effective rate $k_{\mathrm{eff}}$, we now consider the downstream consequences in the reaction network of G$\alpha_{s}$ signaling. We use shorthands for the molecules involved: receptor–ligand complex (RS), G protein (G), adenylate cyclase (AC), adenosine triposphate (ATP) and cyclic adenosine monophosphate cAMP. In this notation, the signaling network (also shown in Figure 1) can be represented in a rather truncated fashion [19,20] by the upstream part
(2)$$\begin{array}{cc} R+S & ↔\mathrm{RS}\\ RS+G & ↔\mathrm{RSG}\\ RSG & \to \mathrm{RS}+G^{*} \end{array}$$
and the downstream part: $$\begin{array}{cc} G^{*}+AC & ↔ACG^{*}\\ ACG^{*}+ATP & \to cAMP+ACG^{*}\\ G^{*} & \to G. \end{array}$$

It is imperative to note that we account for the diffusion-limited reaction by means of an effective first-order reaction law, given by
RS↔RSG.

This replaces Equation (Equation 2) with the previously defined effective rate $k_{\mathrm{eff}}$ for the forward reaction as we move from a single particle-based description to a cell-level concentration language. The concentration of the available G protein is effectively assumed to be constant, as it is encoded in this rate (by way of the parameter *L* in our model of the previous section). Additionally, we are working under the assumption of effectively freely diffusing particles of all species so that interaction is not relevant. We note that nonlinear concentration effects in diffusion-limited reactions have been discussed in the literature [60].

The quantity of interest to us is the concentration [cAMP] of cAMP as a function of external stimulus concentration [S] and the effective rate $k_{\mathrm{eff}}$, which accounts for the experimentally controllable diffusion constant. The cAMP is produced in the downstream part. The connection between the up- and downstream parts of the reaction network is the concentration of activated G protein $\left[G^{*}\right]$. We track the concentrations on the plasma membrane only, so cytosolic and other contributions are not relevant. The activation of G protein is the result of the stimulus–receptor–G protein complex and acts effectively as an enzyme in the downstream part. We will address these two steps individually.

### 3.2. Concentration of Activated G Protein

To determine the dependency of $\left[G^{*}\right]$ on the two control parameters, external stimulus S and dynamics-dependent $k_{\mathrm{eff}}$, we explicitly consider the dynamics of the two relevant compounds involved in the production of activated G protein: $$\begin{array}{cc} \frac{d}{dt}\left[\mathrm{RS}\right] & =k_{\mathrm{RS}}^{+}\left[R\right]\left[S\right]-k_{\mathrm{RS}}^{-}\left[RS\right]-k_{\mathrm{eff}}\left[RS\right]+\left(k_{\mathrm{down}}+k_{\mathrm{RSG}}^{-}\right)\left[\mathrm{RSG}\right]\\ \frac{d}{dt}\left[\mathrm{RSG}\right] & =k_{\mathrm{eff}}\left[\mathrm{RS}\right]-\left(k_{\mathrm{down}}+k_{\mathrm{RSG}}^{-}\right)\left[\mathrm{RSG}\right], \end{array}$$
wherein the last term corresponding to the degradation of RSG is explicitly split up into two contributions: a trivial unbinding with a rate $k_{\mathrm{RSG}}^{-}$ and the downstream conversion (i.e., activation of the G protein and then unbinding) with a rate $k_{\mathrm{down}}$.

We assume dynamics that run for sufficiently long times to be close to equilibrium, corresponding to vanishing time derivatives on the left-hand side of these equations, giving
$$\begin{array}{cc} \left[\mathrm{RS}\right] & =\frac{k_{\mathrm{RS}}^{+}\left[R\right]\left[S\right]+\left(k_{\mathrm{down}}+k_{\mathrm{RSG}}^{-}\right)\left[\mathrm{RSG}\right]}{k_{\mathrm{RS}}^{-}+k_{\mathrm{eff}}}\\ \left[\mathrm{RSG}\right] & =\frac{k_{\mathrm{eff}}\left[\mathrm{RS}\right]}{k_{\mathrm{down}}+k_{\mathrm{RSG}}^{-}}. \end{array}$$

Additionally, the total number of receptor molecules is constant; in other words, we know that
$$\begin{array}{cc} R_{0} & =[R]+[\mathrm{RS}]+[\mathrm{RSG}]=\mathrm{const}. \end{array}$$

Therefore, we find using the shorthand notations
(3)$$C\left(k_{\mathrm{eff}}\right):=\frac{k_{\mathrm{RS}}^{-}}{k_{\mathrm{RS}}^{+}}\frac{k_{\mathrm{down}}+k_{\mathrm{RSG}}^{-}}{k_{\mathrm{down}}+k_{\mathrm{RSG}}^{-}+k_{\mathrm{eff}}}$$
for the relevant concentration scale and $q_{\mathrm{RSG}}\left(k_{\mathrm{eff}}\right):=\frac{k_{\mathrm{eff}}}{k_{\mathrm{down}}+k_{\mathrm{RSG}}^{-}+k_{\mathrm{eff}}}R_{0}$ for an effective rate constant: (4)$$\begin{array}{cc} \left[\mathrm{RS}\right] & =\frac{k_{\mathrm{down}}+k_{\mathrm{RSG}}^{-}}{k_{\mathrm{down}}+k_{\mathrm{RSG}}^{-}+k_{\mathrm{eff}}}\frac{R_{0}\left[S\right]}{C\left(k_{\mathrm{eff}}\right)+\left[S\right]}\\ \left[\mathrm{RSG}\right] & =q_{\mathrm{RSG}}\left(k_{\mathrm{eff}}\right)\frac{\left[S\right]}{C\left(k_{\mathrm{eff}}\right)+\left[S\right]}. \end{array}$$

It is noteworthy that this is not directly the result one would be looking for, as the variable that can be controlled experimentally is not the concentration [S] of unbound ligands available at the membrane but rather the initial concentration of ligands outside the cell. However, the large reservoir of ligands outside the cell and the cell membrane are in direct and continuous exchange throughout the experiment, so the general form of (Equation 4) as well as the dependencies on $k_{\mathrm{eff}}$ hold. We will therefore also use the shape of this result for the concentration of activated G protein that is directly linked to the RSG complex concentration, as the activation flux is given by $k_{\mathrm{down}}\left[\mathrm{RSG}\right]$ and is ultimately balanced (as the G protein acts enzymatic in the other downstream reactions) by the deactivation flux $k_{G^{*}G}\left[G^{*}\right]$. The equilibrium concentration is given by
(5)$$\begin{array}{cc} \left[G^{*}\right] & =G_{\max}\frac{\left[S\right]}{C+[S]} \end{array}$$
with the same concentration scale $C=C(k_{\mathrm{eff}})$ as before and $G_{\max}=G_{\max}\left(k_{\mathrm{eff}}\right)$, with
(6)$$G_{\max}\left(k_{\mathrm{eff}}\right):=\frac{k_{\mathrm{down}}}{k_{G^{*}G}}q_{\mathrm{RSG}}\left(k_{\mathrm{eff}}\right)=\frac{k_{\mathrm{down}}}{k_{G^{*}G}}\frac{k_{\mathrm{eff}}}{k_{\mathrm{down}}+k_{\mathrm{RSG}}^{-}+k_{\mathrm{eff}}}R_{0}$$
being the concentration at stimulus saturation that is independent of [S] but, crucially, is dependent on $k_{\mathrm{eff}}$. Here, it is especially consequential that our description is limited to plasma membrane-bound G protein. By introducing exchange with cytosolic molecules, the relationship between the membrane-bound concentrations [RSG] and $\left[G^{*}\right]$ becomes more involved.

### 3.3. cAMP Production

Following this dynamical effect on the concentration of activated G protein, we are faced with the issue that further downstream molecules are involved, and the description could become very complex. For simplicity, we shall assume here that the most relevant additional molecules, namely adenylate cyclase (AC) and adenosine triphosphate (ATP), are present in sufficient abundance and their concentration will not change throughout the signaling. With these assumptions in place, we can formulate the dynamical equations for the ${\mathrm{ACG}}^{*}$ and cAMP complexes as
$$\begin{array}{cc} \frac{d}{dt}\left[{\mathrm{ACG}}^{*}\right] & =k_{{\mathrm{ACG}}^{*}}^{+}\left[\mathrm{AC}\right]\left[G^{*}\right]-k_{{\mathrm{ACG}}^{*}}^{-}\left[{\mathrm{ACG}}^{*}\right]\\ \frac{d}{dt}\left[\mathrm{cAMP}\right] & =k_{\mathrm{cAMP}}\left[{\mathrm{ACG}}^{*}\right]\left[\mathrm{ATP}\right]. \end{array}$$


Again, we assume equilibrium and can make use of molecule conservation, this time for the AC ($A_{0}=\left[\mathrm{AC}\right]+\left[{\mathrm{ACG}}^{*}\right]$) to find
$$\begin{array}{cc} \left[{\mathrm{ACG}}^{*}\right] & =\frac{A_{0}\left[G^{*}\right]}{C_{\mathrm{cAMP}}+\left[G^{*}\right]}\\ \frac{d}{dt}\left[\mathrm{cAMP}\right] & =q_{\mathrm{cAMP},G^{*}}\frac{\left[G^{*}\right]}{C_{\mathrm{cAMP}}+\left[G^{*}\right]} \end{array}$$
with a rate $q_{\mathrm{cAMP},G^{*}}:=k_{\mathrm{cAMP}}A_{0}\left[\mathrm{ATP}\right]$ and concentration scale $C_{\mathrm{cAMP}}:=k_{{\mathrm{ACG}}^{*}}^{-}/k_{{\mathrm{ACG}}^{*}}^{+}$, in which we regard the [ATP] concentration available as a given parameter. Using our previous result from Equation (Equation 5), we can rephrase this into a form suitable for experimental evaluation using $q_{\mathrm{cAMP},S}:=\frac{q_{\mathrm{cAMP},G^{*}}G_{\max}}{G_{\max}+C_{\mathrm{cAMP}}}$ and $H:=\frac{C_{\mathrm{cAMP}}}{G_{\max}+C_{\mathrm{cAMP}}}C$: (7)$$\begin{array}{cc} \frac{d}{dt}\left[\mathrm{cAMP}\right] & =q_{\mathrm{cAMP},S}\frac{\left[S\right]}{H+[S]}. \end{array}$$

This result implies a prediction for experimentally observable dose–response curves. For our purposes, we need the dependence of such curves on the dynamical parameter $k_{\mathrm{eff}}$ which enters via $G_{\max}$ and *C* defined in Equations (Equation 3) and (Equation 6), respectively. As our reasoning is focused on the diffusion-limited case throughout this perspective, we consider only the leading order in $k_{\mathrm{eff}}/k_{\mathrm{down}}$ (i.e., the relevant parameter dependence is $G_{\max}∼k_{\mathrm{eff}}$, whereas C∼const). With this, we can formulate a reduced version of Equation (Equation 7) that highlights the (leading order) dependence on the dynamically controlled parameter $k_{\mathrm{eff}}$. We introduce a reference scale $k_{0}$ at which the half-width of the Michaelis–Menten-type fraction is $E_{0}$ and find
(8)$$\begin{array}{cc} \frac{d}{dt}\left[\mathrm{cAMP}\right] & \propto \frac{k_{\mathrm{eff}}}{k_{0}}\frac{\left[S\right]}{\frac{2E_{0}}{1+\frac{k_{\mathrm{eff}}}{k_{0}}}+\left[S\right]}. \end{array}$$

Thus, we find that there is a very strong signature of the dynamics in the instantaneous cAMP production, since there is a first-order correction in the deviation $k_{\mathrm{eff}}/k_{0}-1$ to it. However, the initial slopes of the dose–response curves are practically hard to assess, and it is typically more convenient to infer steady state values for cAMP in response to an external stimulus. Obviously, such a steady state is not predicted from the cAMP production rate in Equation (Equation 8), because our considerations neglected the effects of cAMP degradation related, for example, to phosphodiesterases (PDEs) [4]. However, we can progress by knowing that the actual normalized dose–response curve N([S]) has a specific sigmoidal shape in the stimulus concentration. Normalization in this case refers to a rescaling and shifting of the curve such that N([S]→0)=0 and N([S]→∞)=1. The same shape is to be expected as a function of the instantaneous cAMP production rate. We call this rate, given by the right-hand side of Equation (Equation 8), *f* and therefore are lead to approximate the overall gestalt of N[f] by a Hill–Langmuir-type function: $$\begin{array}{c} N\left[f\right]\approx \frac{f^{n}}{u^{n}+f^{n}} \end{array}$$
with an exponent *n* that is usually interpreted as encoding cooperativity (n>1 being positive and n<1 being negative cooperativity) and a constant *u*. Using our earlier result from Equation (Equation 8), and assuming the cAMP production itself to be far from saturation, this leads to
(9)$$\begin{array}{c} N\left(\left[S\right],k_{\mathrm{eff}}\right)\approx \frac{{\left[S\right]}^{n}}{{\left(2F_{0}\right)}^{n}{\left(\frac{k_{\mathrm{eff}}}{k_{0}}\left(1+\frac{k_{\mathrm{eff}}}{k_{0}}\right)\right)}^{-n}+{\left[S\right]}^{n}} \end{array}$$
with a constant $F_{0}$. Thus, we can make a prediction for the scaling of the effective EC50-value of the dose–response curve. For visualization in Figure 6, we use n=1, partially inspired by the preliminary experimental results shown in Figure 7 and also due to the fact that there is no immediate biophysical intuition justifying visible cooperativity. Actual expressions for the remaining constant $F_{0}$ are too model-dependent to be of any value. One possible experimental realization of this scenario is found by implementing a disruptive change in membrane receptor dynamics via actin depolymerization [38] using Latrunculin B (LatB). Preliminary data (see Figure 7) show a sizable shift in the dose–response curve under the addition of isoprenaline in a manner consistent with our reasoning. The qualitative message that can be inferred here is that there is a shift in the dose–response curves when the dynamic properties of the receptors are altered. There are two obvious lines of further investigation here: (1) establish the correlation of the LatB concentration to the degree of actin polymerization (as well as to the diffusional properties of the GPCRs), and (2) understand the possible cell-type dependence of this process.

## 4. Conclusions

In this short perspective, we revisited the concept of diffusion-limited reactions within the context of GPCR-mediated cellular signaling. Signaling shaping (i.e., mechanisms that allow for a greater complexity in signaling than the relatively small number of signaling cascades) is very relevant and topical. Thus far, considerations about receptor statistics in terms of relative abundances have been taking precedence over the respective dynamics. Acknowledging that an essential part of the GPCR signaling machinery hinges on a diffusion-limited reaction provides the foundation for manifestly relating single-particle dynamics and signal shaping or localization. In turn, all the tools for modulating and quantifying particle dynamics become directly relevant for the study and understanding of spatiotemporal signal shaping.

We have offered a simple model description (Section 2.1 and Section 2.2) in the standard particle-based reaction-diffusion language of the Doi model. This model highlights the relevance of diffusive travel times, in particular for the binding of the G protein to the GPCR. This offers an accessible pathway to understanding the genesis of an effective rate for the production of activated G protein that is strongly dependent on (which in the limit of low mobility becomes proportional to) the diffusive constant of the relative motion between a ligand-bound receptor and unactivated G protein. This strong direct relation between the dynamical properties and reaction times is the hallmark of diffusion-limited reaction times. A graphical summary of the signaling modulation ideas discussed can be found in Figure 8.

We have re-established the importance of single-molecule dynamics, the importance of which might not have been fully appreciated in the context of GPCR signaling, and raised some points for future investigation. The first point is, of course, the potential of a direct modulation of the diffusive properties, such as by means of membrane composition or a change in the underlying actin mesh (cp. Figure 3). The second is the role of geometry. As an example, we studied the case of a cylinder where the curvature along the circumferential direction effectively modulated the dimensionality in the diffusive law. Long-time diffusion is essentially one-dimensional on a cylinder. At a constant surface density of the relevant molecules, this means that in the diffusion-limited regime, there is not only a relation of the form k∼D (a reaction rate scaling like the relative diffusion constant) but also $k∼\rho^{\gamma }$ (the reaction rate being geometry-dependent, with ρ being the radius of the cylinder and γ≈2/3 being some exponent) (see Figure 5). Thus, reactions that are diffusion-limited or for which diffusive travel times are important, with GPCR signaling being one of them, are geometry-dependent due to diffusion dynamics alone. A highly curved membrane will generate a coupling of dynamics and geometry that has a profound effect on the reaction kinetics. Notably, this is independent of the changes in energy landscapes that occur, for example, due to changing distances within the lipid bilayer. The quantitative comparison of dynamical effects compared to energetic contributions is one highly interesting route that we would like to point to in this perspective.

We discussed the relevance of the diffusion limitation in the binding to the G protein to the abundance of the second messenger cAMP for a truncated reaction network in Section 3. The key notion here is the possibility of a substantial shift in the typical dose–response curves (Figure 6), which we substantiated with some original preliminary results on the effect of disrupting the actin mesh and thus enhancing receptor mobility, as shown in Figure 7. We shall note here that our analysis is not only deliberately simplified but further limited to the case of $G\alpha_{s}$ protein. While the general ideas are independent of the specific signaling cascade, as they relay to the binding of G protein and the receptor–ligand complex, different Gα subunits would obviously lead to second messengers other than cAMP or to its reduction rather than an increase for the case of $G\alpha_{i}$. Moreover, the network triggered downstream from this will differ, and the effect from modulation of the receptor dynamics will differ as well. Furthermore, we are concentrating in our investigation on membrane-bound signaling actors, neglecting, for example, potential signaling interactions occurring intracellularly [61] or the role of the cytosolic G protein pool [62], which we believe do not affect the generality of our conclusions.

In summary, a quantitative analysis of the relation between receptor dynamics and signaling efficacy in an experimental setting appears to be an important direction for future work. To this end, it would also be beneficial to explore ways of incorporating cAMP responses beyond steady state values.

With the advent of single-molecule imaging and spectroscopy techniques, receptor dynamics are now directly accessible [29,32], making revisiting their importance in the context of signal shaping a timely and important topic.

## Figures and Tables

**Figure 1 cells-11-01660-f001:**
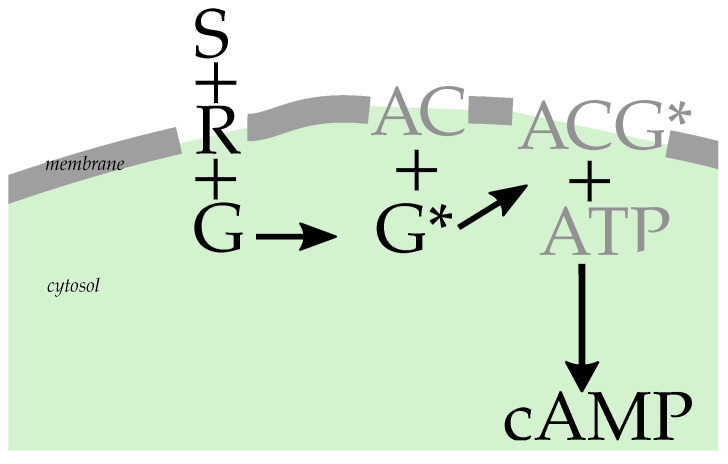
Simplified kinetic model illustrating how the second messenger molecule cAMP is produced downstream. The initial ternary reaction between the ligand (S), receptor (R) and G protein (G) on the cell membrane (gray line) leads to activated G protein (G${}^{*}$). This then is used together with membrane-bound AC and ATP from the cell cytosol (green) for cAMP production. We only depict the steps along the line of the conversion of the extra-cellular signal to production of the second messenger protein cAMP, which we explicitly model in this short perspective.

**Figure 2 cells-11-01660-f002:**
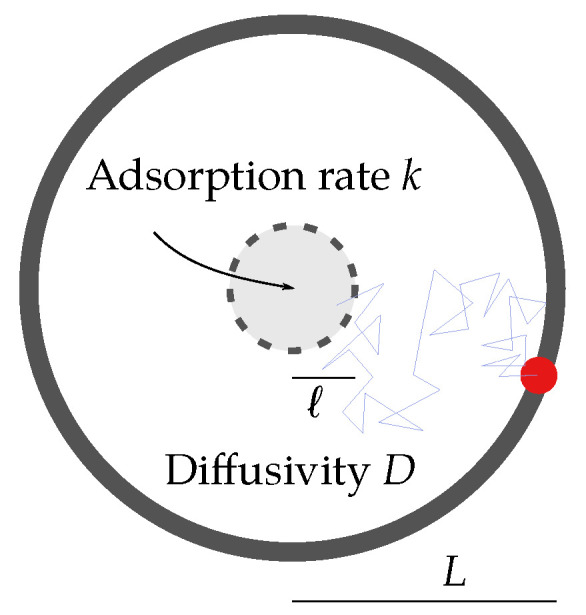
Sketch of the toy model. The particle (red dot) diffuses with a diffusion constant D>0 in a circular domain of radius L>0, searching for the active zone of radius *ℓ* (0<ℓ<L). One example trajectory is depicted by the blue line. Within the active zone, the particle can get absorbed at rate k>0. For the target problem of this work, the particle can be thought of as the ligand-bound receptor RS finding the G protein, but it is moving with the combined diffusion constant.

**Figure 3 cells-11-01660-f003:**
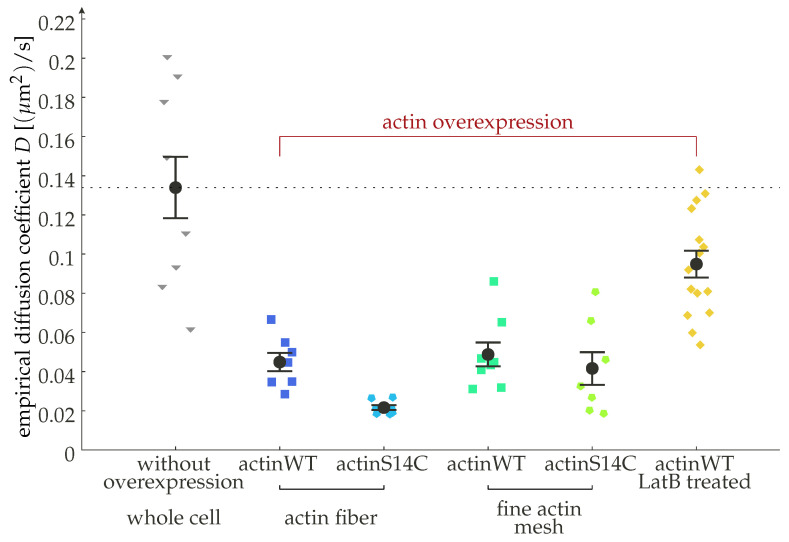
Experimental evidence for how receptor dynamics can be modulated by the subplasmalemmal environment, in particular for the cortical actin mesh, showing the diffusion coefficient of the $\beta_{2}$-adrenergic receptor ($\beta_{2}$-AR) in rat myoblast H9c2 cells for different regions of the cell and conditions. The $\beta_{2}$-AR is significantly slower in cells overexpressing actin. When actin is hyperpolimerized, using the mutant S14C, the diffusion is additionally reduced. The receptors are slowest around large actin bundles (fibers). We added a dashed line corresponding to the wild-type findings to highlight the effect of actin overexpression. Actin depolymerization by Latrunculin B (2 μM) leads to a statistically significant recovery in the diffusion rate of the receptor. Error bars indicate the standard error of the mean. This figure reproduces data from [36].

**Figure 4 cells-11-01660-f004:**
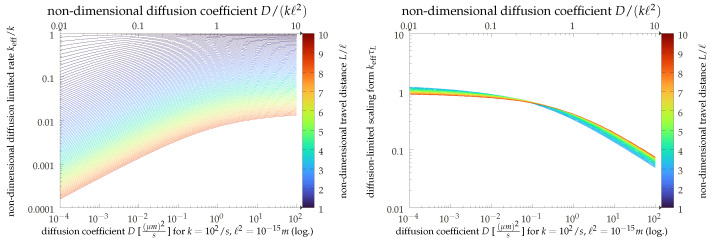
Effective production rate $k_{\mathrm{eff}}$ given by the inverse of the mean first reaction time in the Doi model presented in the main text, with some additional details given in Appendix A. We present the results in dimensionless units (upper ordinate axis) and using exemplary physical values (lower axis). **Left**: Direct results for $k_{\mathrm{eff}}$ as a function of the particle diffusion coefficient for varying ratios of domain size *L* and interaction range *ℓ* as coded by line color. **Right**: The same data, but directly employing the diffusion-limited scaling $k_{\mathrm{eff}}∼\tau_{L}^{-1}$. Here, we show only data for larger values of L/ℓ>3, where the scaling is expected to hold.

**Figure 5 cells-11-01660-f005:**
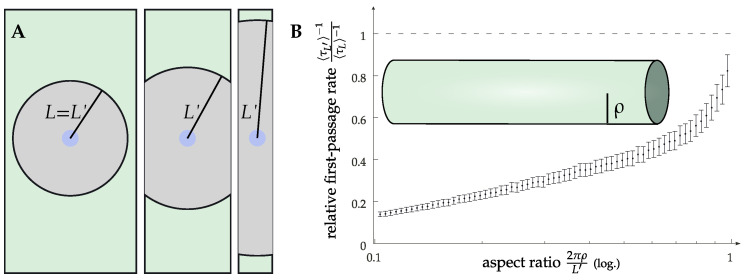
(**A**) Sketch of the change in the Doi model geometry on a cylinder for varying cylinder radii (cylinder is also sketched as an inset of (**B**)). We compare models of the same interaction area $\pi L^{2}$. Due to the periodic boundary conditions, this leads to a larger extension in the axial direction.

**Figure 6 cells-11-01660-f006:**
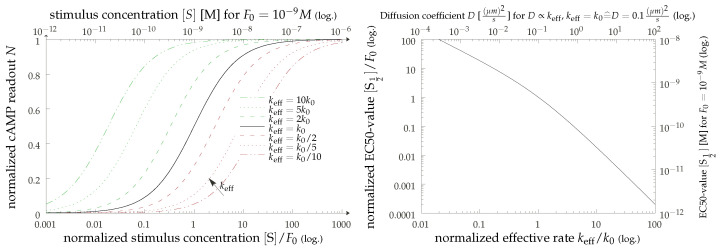
(**Left**) Visualization of the results from Equation (Equation 9). We show artificial dose–response curves for various values of the effective rate $k_{\mathrm{eff}}$ which, in the diffusion-limited regime, corresponds to the relative diffusion constant. In the upper *x*-axis, we provide a possible instance of physical values for a typical value of $F_{0}$. (**Right**) The EC50 values (i.e., the stimulus concentrations for which the response is half maximal as a function of the dynamically controlled effective rate (lower *x*-axis) or the diffusion constant (using typical values; upper *x*-axis)).

**Figure 7 cells-11-01660-f007:**
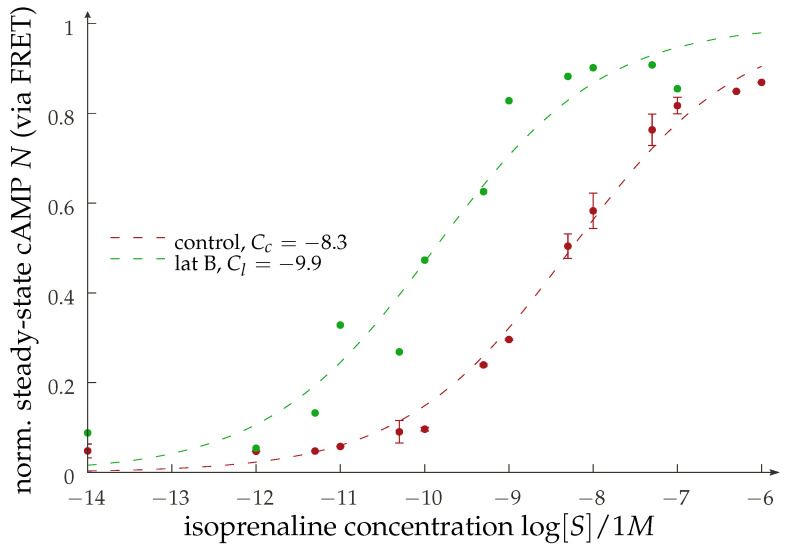
Pilot experiment displaying the dose–response curves for cells stably expressing a FRET cAMP biosensor (Epac1-camps(H187)) and exposed to actin depolymerization. Depicted is the steady state response to an external increasing isoprenaline concentration. The control curve corresponds to two measurements with untreated cells, while the LatB curve is the response with additional LatB (25 nM) and thus a lesser degree of polymerization in the actin. Lower connectivity in the mesh should reduce the dynamical inhibition of the receptors and therefore increase their mobility. The normalization and the fits are performed using a Hill–Langmuir function with n=1. The numerical values for the constant *C* (in M) are given in the plot and quantify the obvious left shift upon the addition of LatB that is qualitatively in line with the reasoning for an increase in mobility.

**Figure 8 cells-11-01660-f008:**
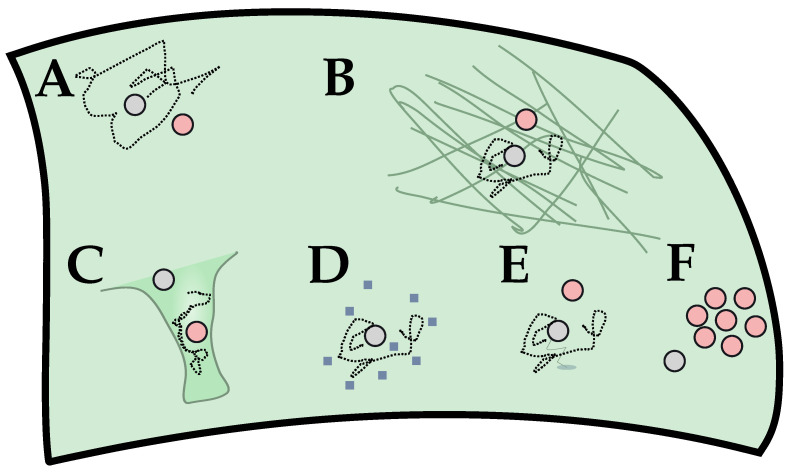
In this perspective, we consider GPCR signaling as a reaction-diffusion process (A). Due to the unavailability of G protein following activation, the diffusive travel times are relevant and can be limiting to the observed reaction rates. From this, local modulation of signaling can be phrased as a local modulation of diffusive properties. We focus on a change in the subplasmalemmal environment (B) through a change in actin expression. This aims to directly modulate the diffusion constant *D* of the relative motion between GPCR and G protein. Less directly, the distribution of diffusive travel time and, thus, the global reaction kinetics can be affected by a change in geometry (i.e., local curvature (C)). The diffusion constant can also be modulated by other means, with possibilities including membrane composition (D) or GIPs binding to the receptor (E). A less direct way for changing diffusion times would be non-trivial spatial statistics (i.e., clustering or oligomerization (F)). The effect of C and F can be partially captured in the simple model discussed here (see Section 2.3). They also are related to the issue of anomalous diffusion (Section 2.3).

## Data Availability

All data and code are available on request.

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
