# Peer review of "The Impact of Membrane Protein Diffusion on GPCR Signaling"

_cells, 2022, doi:10.3390/cells11101660_

Round 1

Reviewer 1 Report

The perspective of Boltz et al addresses an interesting current topic in GPCR research. Aspects of GPCR mobility are calculated and discussed with different views and membrane models. Effects of few further events important in GPCR function such as GPCR oligomerization and larger rigid membrane structures (rafts) should be commented and included in the calculations/discussions. As the models are based on Gs-types of G-proteins, comments on how, or whether, the conclusions will be modified by transfer to other G-protein types should be given.

The final conclusion would clearly benefit if an overview picture illustrating the discussed diffusion mechanisms, restrictions and options of regulation would be given.

Position of Fig. 2, 3 should be changed and aligned with their discussion in the text.

Line 199: ..wide range in reported is…

Line 468: the the respective..

Reviewer 2 Report

I am not a mathematician, so my comments are largely from biological perspective. The authors describe one possible approach to modeling GPCR signaling. While they talk about compartmentalization inside the cell, which is likely biologically important, their model does not contain this feature. Some features of the model appear to be unjustifiable oversimplifications. For example, they don’t take into account activation-dependent changes in G protein distribution within the cell. G proteins are in equilibrium between membrane-associated and cytoplasmic. Inactive heterotrimeric G proteins have two membrane anchors, whereas activated Ga and Gbg have only one membrane anchor each. Thus, activation makes G proteins more likely not to be membrane-associated. The effector enzyme considered here is adenylyl cyclase. It is an integral membrane protein, like GPCRs, so activated Ga interactions with it would have the same complications as inactive heterotrimer interactions with GPCRs.

In addition, editing to eliminate typos and correct the grammar is needed. A few examples: abstract, line 6, “separations” should be “separation”; line 20, “relative low” should be “relatively low”; lines 41-42, as is, the sentence “Eventually, if the reverse reaction is not carried over (by hydrolizing GTP to GDP) the pool of active G proteins would deplete” appears to mean the opposite of what the authors intend; line 44, correct term is “well stirred”, not “well-mixed”; lines 442-443, “cooperation” should be “cooperativity”.

Round 2

Reviewer 2 Report

The manuscript was improved in revision, but several concerns remain.

  1. Within the context of considered model there is no reason to use the term “compartmentalization”. So, rather than adding “song and dance” in the intro, the authors should just drop this term.
  2. The model still ignores the fact that G proteins are in equilibrium between membrane-bound and cytoplasmic form, and this equilibrium is shifted to cytoplasmic form upon G protein activation. The authors do not necessarily need to include this into their deliberately simplified model, but they should acknowledge this fact.
  3. Lines 259-262. In fact, whether and how rhodopsin is organized in disc membranes in vivo is an open question. Rows of rhodopsin molecules were detected on mica support, whereas signaling occurs on discs within rod outer segments. So far, no mathematical model of rhodopsin signaling was appreciably improved by the assumption that rhodopsin is organized/clustered, rather than freely diffusing in the disc membrane. In fact, both super-stoichiometric phosphorylation of rhodopsin discovered long ago (J Biol Chem. 1990 Sep 5;265(25):15333-40; J Biol Chem. 1996 Aug 16;271(33):19826-30) and phosphorylation of non-activated cone pigment co-expressed with rhodopsin in the same rods (J Biol Chem. 2005 Dec 16;280(50):41184-91) are compatible with free diffusion of photopigments and are hard to explain if they are organized in rows observed on mica. As far as other GPCRs are concerned, direct data show that at least some do exist in a monomer-dimer equilibrium (Proc Natl Acad Sci USA 2010 Feb 9;107(6):2693-8; J Cell Biol. 2011 Feb 7;192(3):463-80; Cell Biochem Biophys. 2018 Jun;76(1-2):29-37), but both monomers and dimers are extremely short-lived. The time scale of GPCR-GPCR interactions is comparable to that of GPCR-G protein interactions. That might affect the kinetics of signaling, but the model presented does not include this aspect of GPCR function.
